# Risk Factor Analysis of Early-Onset Cataracts in Taiwan

**DOI:** 10.3390/jcm11092374

**Published:** 2022-04-23

**Authors:** Lung-Hui Tsai, Ching-Chung Chen, Chien-Ju Lin, Sheng-Pei Lin, Ching-Ying Cheng, Hsi-Pao Hsieh

**Affiliations:** 1Department of Optometry, Chung Shan Medical University, Taichung 402, Taiwan; x8c032@yahoo.com.tw (L.-H.T.); haha310201@gmail.com (C.-J.L.); allen0339@gmail.com (S.-P.L.); 2Department of Ophthalmology, Chung Shan Medical University Hospital, Taichung 402, Taiwan; 3Department of Optometry, Asia University, Taichung 402, Taiwan; art0802@asia.edu.tw; 4Department of Special Education, National Taiwan Normal University, Taipei 106, Taiwan

**Keywords:** early-onset cataract, ROS analysis, Depression Anxiety Stress Scales

## Abstract

Purpose: According to previous studies, the prevalence rate of cataracts has increased in recent years. This study aims to investigate and analyze the risk factors of early-onset cataracts in Taiwan. Methods: A total of 71 subjects aged between 20 and 55 were diagnosed with cataracts in a medical center. Participants were divided into three groups: control, early-onset cataract (EOC), and combined (EOC combined with dry eye) groups. Eye examinations including autorefraction, best-corrected visual acuity (BCVA), subjective refraction, axial length, fundus, slit lamp, and reactive oxygen species (ROS, including total antioxidative capacity, TAC; C-reactive protein, CRP; and glutathione peroxidase, GPx) were performed. In addition, a questionnaire on patient information, history, habits, family history, and Depression Anxiety Stress Scales (DASS) was completed before the examination. Results: 27 non-EOC (control group), 20 EOC, and 24 combined patients participated in the study. Compared with the control group, Body Mass Index (BMI), gender, educational level, hypertension, diabetes, hyperlipidemia, chronic pain, and body-related diseases were significantly different between the three groups. Family history was also significantly different: family heart disease, hypertension, asthma, allergies, stroke, and immune system were also significantly different. In addition, subjects who took hypertensive drugs, antihistamines, and other medications were also significantly different. Statistical analysis indicated that best corrective visual acuity and the spherical equivalent were significantly different between the three groups. Similar results were found in CRP blood analysis. Discussion and Conclusion: According to the results, EOC may result from systemic diseases. The risk corresponded to an increase in ROS blood analysis. Furthermore, eye drops and medicine intake significantly influenced EOC patients. To prevent or defer early-onset cataracts, monitoring physical health, CRP, and GPx analysis may be worth considering in the future.

## 1. Introduction

Early-onset cataracts (EOCs) develop between the ages of 20 and 55 years old [1,2]. According to the Ministry of Health and Welfare statistics in 2018, cataracts are the most common eye disease in Taiwan’s ophthalmology clinic, with age-related cataracts accounting for 90.7% of cases, whereas the EOC group accounts for 9.07% [3,4]. The incidence of EOC increases year by year, with a younger age at diagnosis phenomenon [5,6]. Therefore, prevention and incidence reduction of EOC will be an important topic. Studies have found that the risk of complications such as Body Mass Index (BMI) [7], smoking [8,9], and alcohol consumption [10], hypertension, diabetes, hyperlipidemia, chronic obstructive pulmonary, asthma [11], stroke [12], ischemic heart disease [13], hypoparathyroidism [14], high myopia [15], and family disease [16] are associated with a higher risk of EOCs. In addition, research showed that people with EOCs increased their cumulative risk of all cancers, head and neck cancer, liver cancer, and breast cancer with increasing time of disease [17]. EOCs may not be a simple eye disease, such as the imbalance between the antioxidant capacity and oxidative stress, inflammation, metabolic-related syndromes, and genetic factors [18,19,20]. The above literature indicates that EOCs may require more attention than age-related cataracts.

Clinical experience and research results have also found that cataracts can cause a decline in functional vision, including contrast sensitivity, visual acuity, glare, photophobia, diplopia, and visual field, and can also lead to a decline in quality of life, occupation, academic performance, bicycle riding, behavioral performance, even mental or psychological state [21,22,23]. To sum up, cataracts are not only common in the elderly; cataracts tend to be in younger patients. Therefore, the purpose of this study was to investigate the possible risk factors of EOC.

## 2. Materials and Methods

A cross-sectional study was conducted from 11 November 2019 to 30 March 2020 in the Department of Ophthalmology, Cheng Ching Hospital, Taichung. All the procedures were in accordance with the Declaration of Helsinki. Approval was obtained from the Institutional Review Board of the Chung Shan Medical University Hospital (Taichung, Taiwan) (Approval number: CS18131). Due to the physical, spirit, and compliance, each subject may have taken several attempts to complete the examination. STROBE guidelines were used for reporting the manuscript [24,25].

### 2.1. Research Subjects

To investigate the risk factors associated with EOC, patients were all diagnosed by the same ophthalmologist between the ages of 20 and 55. Patients with congenital cataracts or other eye diseases or those unable to cooperate with this study were excluded. A total of 100 adults participated initially; 29 subjects were later excluded, one had a congenital cataract, five had experienced myopia surgery or retinal surgery, three had age-related degeneration, two had diabetic retinal disease, and 18 others dropped out or did not finish all examinations. Because a relatively high proportion of Taiwanese have dry eye syndrome, this study divided the patients with cataracts only into one group, and the patients with cataracts who were also diagnosed as early-stage dry eye syndrome by doctors as another group. The final total number of participants in this study was 71. Subjects were divided into three groups according to the ophthalmologist’s diagnosis, 27 non-EOC (control group), 20 EOC, and 24 combined group (EOC combined with dry eye) participated in the study. The ages of the three groups were significantly different (F = 3.76, *p* = 0.028), Tukey HDS comparison indicated that the EOC and combined groups were significantly older than the control group. Although the ages of the three groups of subjects were all within the age defined by early-onset cataracts, there was still an age gap, thus analysis must be conducted to control for age.

### 2.2. Research Materials

Variances in the study including autorefractor (NIDEK ARK-510A), objective refraction, distance visual acuity, non-contact intraocular pressure (IOP, non-contact Tonopachy NIDKE NT-530P), axial length (AL-Scan NIDEK 230488), slit lamp (TOPCON SL-7F), fundus and Optical Coherence Tomography (OCT. NIDEK RS-3000). In addition, total antioxidant blood included glutathione peroxidase (GPx), total antioxidative capacity (TAC), and C-reactive protein (CRP) were measured. Before the examination, a questionnaire was completed detailing each patient’s basic information, history, habit, family history, and Depression Anxiety Stress Scales (DASS).

### 2.3. Data Analysis and Statistical Analysis

The sample size of this study was determined using G*Power analysis, under effect size d = 0.5, α = 0.05, power (1-β) = 0.90. The calculated results of the total sample size were 70. All data were performed and analyzed using SPSS 22.0 statistical software (IBM, Armonk, NY, USA). A value of *p* < 0.05 was considered statistically significant. One-way ANOVA, Pearson χ^2^, and multi-logistic regression analyses were performed.

## 3. Results

To investigate the risk factors associated with EOC, subjects were all diagnosed by the same ophthalmologist between the ages of 20 and 55. The final effective number of participants was 71, and subjects were divided into three groups (27 control group, 20 EOC, and 24 combined group, Table 1) according to the ophthalmologist’s diagnosis. One-way ANOVA analysis indicated that the ages of the three groups were significantly different. The EOC group and the combined group were significantly older than the control group, following analysis must be conducted under controlling for age.

### 3.1. Background Possibility Risk Factors of EOC

#### 3.1.1. Background Comparison between Groups

ANOVA analysis indicated no significant differences in participants’ height (F = 1.53, *p* = 0.22) or weight (F = 0.41, *p* = 0.66). However, Pearson χ^2^ analysis showed that gender (χ^2^ = 8.08, *p* = 0.04), BMI (χ^2^ = 6.89, *p* = 0.032), and educational level (χ^2^ = 6.33, *p* = 0.05) was significantly different between each group; females, those with a higher BMI, and those with a higher education level exhibited an increased risk of EOC.

#### 3.1.2. Family History Comparison between Three Groups

Pearson χ^2^ analysis showed that family heart disease (χ^2^ = 6.06, *p* = 0.05), family hypertension (χ^2^ = 9.07, *p* = 0.01), family asthma (χ^2^ = 5.25, *p* = 0.04), family stroke (χ^2^ = 5.04, *p* = 0.04), familial immune system disease (χ^2^ = 4.44, *p* = 0.05), and family allergies (χ^2^ = 3.19, *p* = 0.02) were significantly difference between all three groups; the percentage in the EOC and combined groups were significantly higher than that in the control group. However, all familial eye diseases, such as family cataract, family glaucoma, and family diabetes did not reach statistical significance, as shown in Figure 1.

#### 3.1.3. Healthy Status Comparison between Each Group

Pearson χ^2^ analysis showed that the subjects themselves who suffered from hypertension (χ^2^ = 10.50, *p* = 0.00), diabetes (χ^2^ = 4.80, *p* = 0.05), hyperlipidemia (χ^2^ = 5.25, *p* = 0.04), high myopia (χ^2^ = 4.67, *p* = 0.05), chronic pain (χ^2^ = 5.01, *p* = 0.04), and other illnesses (χ^2^ = 6.30, *p* = 0.04) were significantly different, the percentage in the EOC and combined groups were significantly higher than that in the control group; no statistically significant differences were determined in other diseases among the three groups (Table 2).

#### 3.1.4. Lifestyle Habits and Drug Use Comparisons between Each Group

The results of the Pearson χ^2^ test for the lifestyle habits survey showed that there was no significant difference in smoking (χ^2^ = 3.816, *p* = 0.43), alcohol (χ^2^ = 0.686, *p* = 0.71), regular exercise (χ^2^ = 2.027, *p* = 0.36), use of mobile and computer (χ^2^ = 2.222, *p* = 0.33), coffee (χ^2^ = 2.379, *p* = 0.30), tea (χ^2^ = 0.112, *p* = 0.95), cola, or other refreshing drink (χ^2^ = 1.004, *p* = 0.60).

Pearson χ^2^ test of drug use also showed no significant differences in steroid (χ^2^ = 1.291, *p* = 0.53), amiodarone (χ^2^ = 2.586, *p* = 0.27), hormone (χ^2^ = 0.817, *p* = 0.67), or analgesics (χ^2^ = 2.523, *p* = 0.28); however, other medications, such as blood pressure (χ^2^ = 10.420, *p* = 0.00), antihistamines (χ^2^ = 4.804, *p* = 0.05), and other drugs (χ^2^ = 9.991, *p* = 0.01), were significantly different between the three groups. The EOC and combined groups received the higher proportion of anti-hypertensive, antihistamine, and other drugs (Table 3).

### 3.2. Psychological Possibility Risk Factors of EOC

The DASS questionnaire was used to determine subjects that had experienced psychological problems. Pearson χ^2^ analysis showed that the subjects diagnosed as EOC or combined group had significantly higher levels of anxiety (χ^2^ = 18.524, *p* = 0.018) and stress (χ^2^ = 20.368, *p* = 0.002) than the control group. At the same time, depression (χ^2^ = 8.563, *p* = 0.380) was insignificant (Figure 2).

### 3.3. Blood Possibility Risk Factors of EOC

Blood analysis determined that there were no significant differences in GPx (χ^2^ = 1.267, *p* = 0.53) or TAC (χ^2^= 1.512, *p* = 0.47); while CRP index (χ^2^ = 7.856, *p* = 0.02) showed a significant difference between all three groups; the percentage in the EOC and combined groups were much higher than that in the control group, as shown in Table 4. A C-reactive protein (CRP) test measured the level of C-reactive protein in blood. CRP is a protein made by the liver and excreted into the bloodstream in response to inflammation which may protect tissues during injury or infection.

### 3.4. Visual Function Possibility Risk Factors of EOC

Visual function examination included refractive errors, axial length, cup to disc ratio (CD ratio), intraocular pressure (IOP), and visual acuity. Since the data from the left and right eyes are not significantly different, the data of ocular physiology are mainly from the right eye. ANOVA analysis showed significant differences in best correct visual acuity (F = 22.95, *p* = 0.00) and spherical equivalent (F = 3.26, *p* = 0.04), but did not show significant difference in astigmatic (F = 0.24, *p* = 0.79), IOP (F = 0.946, *p* = 0.393), CD ratio (F = 0.308, *p* = 0.74), or axial length (F = 2.24, *p* = 0.12).

In summary, (1) family disease, such as family heart disease, hypertension, asthma, stroke, immune system diseases, and allergies; (2) patients’ background, BMI, gender, education level, best corrective visual acuity (BCVA), spherical equivalent; or (3) patients themselves suffered from hypertension, diabetes, hyperlipidemia, and chronic pain; drug use for blood pressure, antihistamines, anxiety, stress, or even the detection of CRP index in the blood are helpful to detect EOC, which is a topic worthy of discussion.

### 3.5. Linear Regression Analysis on Predicting EOC

The risk factor analysis of patients with EOC in Taiwan was based on the basic information of the research subjects, health status survey, living habits survey, anxiety scale, eye examination, and blood analysis that had shown significant differences using ANOVA and Chi-square analysis as independent variables. In addition, the preoperative group was subjected to multinomial logistic regression analysis as the dependent variable to predict the risk factors of EOC with higher explanatory power (Table 5 and Table 6).

The multinomial logistic regression analysis showed that for the background variables (χ^2^ = 9.889, *p* = 0.042, Cox R^2^ = 0.125), family disease (χ^2^ = 35.470, *p* = 0.003, Cox R^2^ = 0.341), healthy status (χ^2^ = 48.972, *p*= 0.00, Cox R^2^= 0.438), medication (χ^2^ = 39.619, *p* = 0.001, Cox R^2^ = 0.373), psychology (χ^2^ = 13.510, *p*= 0.009, Cox R^2^ = 0.149), blood (χ^2^ = 7.452, *p*= 0.024, Cox R^2^ = 0.170), and eye examination (χ^2^ = 18.685, *p* = 0.001, Cox R^2^ = 0.215), each dimension had good and significant predictive ability for EOC. The overall variable can explain up to 88.2% of EOC (χ^2^ = 72.990, *p* = 0.004, Cox R^2^ = 0.883). Strong predictors after screening included BMI, educational level, family heart disease, family hypertension, family allergies, high myopia, other illness, other drugs taken, stress, CRP, BCVA, and spherical equivalent. For example, while the stress index increased by one unit, the risk of developing EOC was 3.258 higher than the control group.

## 4. Discussion

Among the background data, the ages of the three groups were significantly different. The EOC group and combined group were significantly older than the control group. In addition, a higher BMI value, females, and highly educated subjects represented a higher proportion of those suffering from EOC [26,27]. Most participants were excluded at the beginning due to being diagnosed with eye-related diseases or having experienced eye-related surgery. Patients with eye-related diseases or having experienced eye-related surgery had a high proportion of EOC, among which retinal diseases and retinal surgery have the greatest impact. In the family history, relatives with heart disease, hypertension, asthma, stroke, immune system diseases, and allergic constitution have a high proportion of EOC, among which family heart disease, family hypertension, and family allergies have the greatest impact [28,29]. The physiological mechanism of genes, genetic inheritance, environment, and diet remains to be clarified.

In the health survey of the subjects themselves, the patients themselves were suffering from hypertension, diabetes, hyperlipidemia, high myopia, chronic pain, and abnormal CRP index in the blood analysis. A higher proportion of patients with EOC had the most significant impact on other physical illnesses and CRP values [1,30,31,32,33,34,35]. It is reasonable that people with a high BMI are more likely to have symptoms such as high blood sugar, blood pressure, and high blood lipids, so the risk of developing cataracts is higher than that of the normal BMI group. However, there have been no reports to determine whether monitoring the C-reactive protein index is helpful to delay or detect EOC [32,33,34,35]; a relevant topic for future discussion.

In terms of medication, patients with EOC used a high proportion of blood pressure lowering drugs, antihistamines, and other drugs [26,27]. It is worth noting that a high proportion of patients with EOC take antihistamine drugs, which may cause oxidative pressure due to long-term inflammation of the body, which may indirectly lead to the formation of cataracts [28,29].

In examining ophthalmology and optometry, the best corrective visual acuity, spherical equivalent, has a high proportion of patients with early-onset cataracts, and the BCVA before cataract surgery has the greatest impact [36,37]. Although the CD ratio did not show significant differences between groups, previous literature indicated a high proportion of patients with EOC have abnormal CD values, which is related to patients which have also been diagnosed with glaucoma. According to Law and Wang [38], glaucoma is complicated by cataracts, and may compress the optic nerve to cause abnormal CD values. However, at present, the relationship between glaucoma and EOC has yet to be directly linked.

Psychological stress and anxiety are also risk factors for EOC, and the ability of stress to predict EOC is noteworthy. However, there is very little relevant literature to directly point out whether the stress variable in the anxiety scale can be applied to the detection of EOC or whether it will improve the incidence of EOC [23]; a topic worthy of discussion in the future.

## 5. Conclusions

This study found that BMI, gender, education level, preoperative visual acuity, family history of heart disease, hypertension, asthma, apoplexy, immune system diseases, and allergic constitution, and people themselves having hypertension, diabetes, hyperlipidemia, and chronic pain disorders, in addition to taking anti-hypertensive medicine and antihistamines had a high proportion of EOCs. Furthermore, the detection of CRP index in blood, BCVA, and myopia control helped detect EOC, which is also an important topic worthy of follow-up discussion. Although, as mentioned above, the physiological mechanism of genes, genetic inheritance, environment, and diet remains to be clarified, it is suggested that in the future, the government should support the funds to the ophthalmic medical association and should be expanded to carry out research in various regions of the country with more research samples to obtain more research results, and carry out a synchronous analysis with the health insurance database, which will be the research direction in the future.

## Figures and Tables

**Figure 1 jcm-11-02374-f001:**
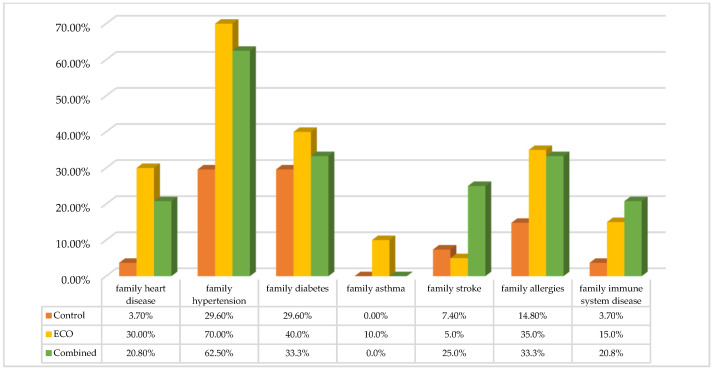
Family history disease prevalence between each group under controlling of age variance.

**Figure 2 jcm-11-02374-f002:**
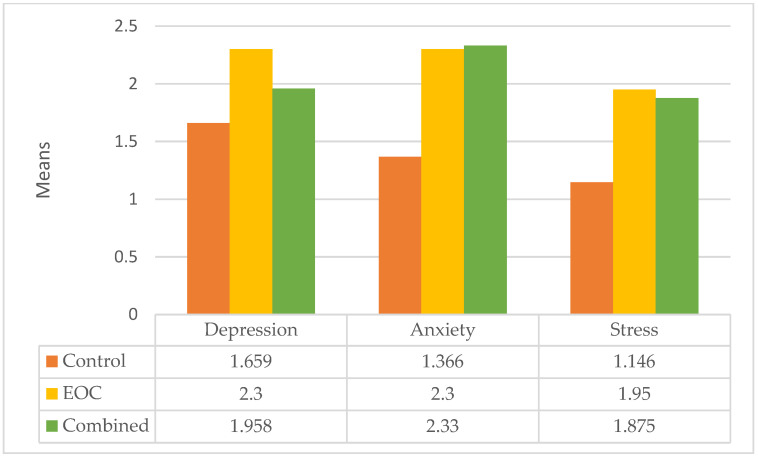
Psychological status of DASS-42 between each group.

**Table 1 jcm-11-02374-t001:** Patient gender and age characteristics for each experimental group.

	Number	%	Gender	Number	Average of Age
Control group	27	38.0%	Male	12	45.17 ± 9.722
			Female	15	44.60 ± 4.388
Early-onset cataract (EOC)	20	28.2%	Male	12	49.92 ± 3.679
			Female	8	49.75 ± 4.464
Combined group (EOC + dry eye)	24	33.8%	Male	9	49.33 ± 3.536
			Female	15	49.40 ± 6.390

**Table 2 jcm-11-02374-t002:** Health status analysis using Pearson χ^2^ for each group.

Controlling Age Variance		Groups			Pearson χ^2^	*p*
		Normal	Cataract	Combined		
Hypertension	yes	0	7	6	10.495	0.00 **
		0.0%	35.0%	25.0%		
	no	27	13	18		
		100.0%	65.0%	75.0%		
Diabetes	yes	0	3	4	4.799	0.05 *
		0.0%	15.0%	16.7%		
	no	27	17	20		
		100.0%	85.0%	83.3%		
Asthma	yes	1	1	1	0.048	0.98
		3.7%	5.0%	4.2%		
	no	26	19	23		
		96.3%	95.0%	95.8%		
Hyperlipidemia	yes	0	2	0	5.248	0.04 *
		0.0%	10.0%	0.0%		
	no	27	18	24		
		100.0%	90.0%	100.0%		
Immune system disease	yes	1	1	1	0.048	0.98
		3.7%	5.0%	4.2%		
	no	26	19	23		
		96.3%	95.0%	95.8%		
Cancer	yes	1	0	1	0.817	0.66
		3.7%	0.0%	4.2%		
	no	26	20	23		
		96.3%	100.0%	95.8%		
High myopia	yes	10	12	8	3.665	0.16
		37.0%	60.0%	33.3%		
	no	17	8	16		
		63.0%	40.0%	66.7%		
High astigmatism	yes	1	2	0	2.725	0.26
		3.7%	10.0%	0.0%		
	no	26	18	24		
		96.3%	90.0%	100.0%		
Thyroid dysfunction	yes	1	2	2	0.788	0.67
		3.7%	10.0%	8.3%		
	no	26	18	22		
		96.3%	90.0%	91.7%		
Galactosemia	yes	0	1	0	2.586	0.27
		0.0%	5.0%	0.0%		
	no	27	19	24		
		100.0%	95.0%	100.0%		
Homo cystinuria	yes	0	0	1	1.986	0.37
		0.0%	0.0%	4.2%		
	no	27	20	23		
		100.0%	100.0%	95.8%		
Migraine	yes	3	4	6	1.692	0.43
		11.1%	20.0%	25.0%		
	no	24	16	18		
		88.9%	80.0%	75.0%		
Irritable Bowel Disorder	yes	2	0	2	1.683	0.43
		7.4%	0.0%	8.3%		
	no	25	20	22		
		92.6%	100.0%	91.7%		
Chronic pain	yes	0	3	1	5.009	0.04 *
		0.0%	15.0%	4.2%		
	no	27	17	23		
		100.0%	85.0%	95.8%		
Head injury	yes	1	2	1	1.004	0.61
		3.7%	10.0%	4.2%		
	no	26	18	23		
		96.3%	90.0%	95.8%		
Other illnesses	yes	3	8	9	6.299	0.04 *
		11.1%	40.0%	37.5%		
	no	24	12	15		
		88.9%	60.0%	62.5%		

* *p* < 0.05, ** *p* < 0.01.

**Table 3 jcm-11-02374-t003:** Pearson χ^2^ analysis on the health status between groups.

Controlling Age Variance		Groups			Pearson χ^2^	*p*
		Normal	Cataract	Combined		
Steroid	yes	1	2	2	1.291	0.53
		3.7%	10.0%	8.3%		
	no	26	18	22		
		96.30%	90.00%	91.70%		
Anti-hypertensive drug pressure	yes	0	7	5	10.420	0.00 **
		0.0%	35.0%	20.8%		
	no	27	13	19		
		100.0%	65.00%	79.20%		
Amiodarone	yes	0	1	0	2.586	0.27
		0.0%	5.0%	0.0%		
	no	27	19	24		
		100.0%	95.0%	100.0%		
Antihistamine	yes	1	4	1	4.804	0.05
		3.7%	20.0%	4.2%		
	no	26	16	23		
		96.3%	80.0%	95.8%		
Hormone therapy	yes	1	0	1	0.817	0.67
		3.7%	0.0%	4.2%		
	no	26	20	23		
		96.3%	100.0%	95.8%		
Painkiller	yes	1	3	4	2.523	0.28
		3.7%	15.0%	16.7%		
	no	26	17	20		
		96.3%	85.0%	83.3%		
Other drugs	yes	4	7	6	9.991	0.01 *
		14.8%	35.0%	25.0%		
	no	23	13	18		
		85.2%	65.0%	75.0%		

* *p* < 0.05, ** *p* < 0.01.

**Table 4 jcm-11-02374-t004:** Pearson χ^2^ analysis of blood results between each group.

Controlling Age Variance		Groups			Pearson χ^2^	*p*
		Normal	Cataract	Combined		
GPx index	normal	12	7	13	1.267	0.53
		80.0%	77.8%	92.9%		
	abnormal	3	2	1		
		20.0%	22.2%	7.1%		
CRP index	normal	15	7	8	7.856	0.02 *
		100.0%	70.0%	57.1%		
	abnormal	0	3	6		
		0.0%	30.0%	42.9%		
TAC index	normal	10	7	12	1.512	0.47
		66.7%	70.0%	85.7%		
	abnormal	5	3	2		
		33.3%	30.0%	14.3%		

* *p* < 0.05; total antioxidative capacity, TAC; C-reactive protein, CRP; glutathione peroxidase, GPx.

**Table 5 jcm-11-02374-t005:** Multinomial logistic regression analysis in each dimension (controlling age variance).

Dimension	χ^2^	*p*	Cox R^2^
Background	9.889	0.042	0.125
Family disease	35.470	0.003	0.341
Healthy status	48.972	0.000	0.438
Medication	39.619	0.001	0.373
Psychology	13.510	0.009	0.149
Blood	7.452	0.024	0.170
Eye examination	18.685	0.001	0.215
Overall	72.990	0.004	0.883

**Table 6 jcm-11-02374-t006:** Multinomial logistic regression analysis between each group.

		*B*	S.E.	Wald	*p*-Value	Exp(B)
BMI	EOC vs. Control	−1.043	0.606	2.961	0.085	0.352
	Combined vs. Control	−1.873	0.648	8.357	0.004 **	0.154
Educational level	EOC vs. Control	−1.456	0.733	3.949	0.047 *	0.233
	Combined vs. Control	−1.755	0.730	5.785	0.016 *	0.173
Family heart disease	EOC vs. Control	−2.165	0.987	4.812	0.028 *	0.115
	Combined vs. Control	−1.456	1.025	2.018	0.155	0.233
Family hypertension	EOC vs. Control	−1.420	0.651	4.760	0.029 *	0.242
	Combined vs. Control	−0.780	0.620	1.580	0.209	0.459
Family allergies	EOC vs. Control	−1.133	0.772	2.155	0.142	0.322
	Combined vs. Control	−1.679	0.775	4.699	0.030 *	0.187
High myopia	EOC vs. Control	−1.950	0.758	6.625	0.010 *	0.142
	Combined vs. Control	−1.442	0.616	5.515	0.019 *	0.143
Other illness	EOC vs. Control	−1.900	0.830	5.235	0.022 *	0.150
	Combined vs. Control	−1.697	0.732	5.367	0.021 *	0.183
Other drugs	EOC vs. Control	−1.677	0.880	3.635	0.057	0.187
	Combined vs. Control	−1.303	0.795	2.684	0.101	0.272
Stress	EOC vs. Control	1.182	0.504	5.505	0.019 *	3.259
	Combined vs. Control	0.850	0.480	3.131	0.077	2.339
CRP	EOC vs. Control	6.307	3.689	2.923	0.087	548.43
	Combined vs. Control	6.337	3.657	3.003	0.083	565.38
BCVA	EOC vs. Control	−12.207	5.620	4.718	0.030 *	0.005
	Combined vs. Control	−7.811	5.994	1.698	0.193	0.033
Spherical equivalent	EOC vs. Control	0.339	0.149	5.181	0.023 *	1.404
	Combined vs. Control	0.242	0.106	5.202	0.023 *	1.274

* *p* < 0.05, ** *p* < 0.01; body mass index (BMI); C-reactive protein (CRP); best corrective visual acuity (BCVA).

## Data Availability

The datasets used during the current study are available from the corresponding author.

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
