# Peer review of "Risk Factor Analysis of Early-Onset Cataracts in Taiwan"

_jcm, 2022, doi:10.3390/jcm11092374_

Round 1

Reviewer 1 Report

Improve English, some paragraphs are hardly understandable.

If EOC group and the combined group were significantly older than the control group, the groups were not comparable. Are the statistics still valid?

What is the rationale of including an EOC + Dry eye group? Comparison between those  gruops is not clear in table 5.

Simplify, make more understandable and visual tables 2, 3, 4 and 5.

Check acronyms (must be introduced when first used and then be systematic, even common ophthalmological terms as IOP and CD ratio)

There are misspellings and repeated phrases in the text

Author Response

  1. Improve English, some paragraphs are hardly understandable.It has been sent to the editing company for revision
  2. If EOC group and the combined group were significantly older than the control group, the groups were not comparable. Are the statistics still valid?                                                                                                                    Line 91-93  Although the ages of the three groups of subjects were all within the age defined by early-onset cataracts, there is still an age gap, following analysis must be conducted under controlling for age
  3. What is the rationale of including an EOC + Dry eye group? Comparison between those  gruops is not clear in table 5.                                                   Line 83-86 Because a relatively high proportion of Taiwanese have dry eye syndrome, this study divided the patients with cataracts only into one group, and the patients with cataracts who were also diagnosed as early stage dry eye syndrome by doctors as another group.
  4. Simplify, make more understandable and visual tables 2, 3, 4 and 5. Revised, see Line3 191-192, Line199-200, Line 278-279, add one Table prior   to Table 5. 
  5. Check acronyms (must be introduced when first used and then be systematic, even common ophthalmological terms as IOP and CD ratio) Revised, see Line 160, 287-288
  6. There are misspellings and repeated phrases in the text                                           Revised

Reviewer 2 Report

Dear author(s),

Thanks for your submission on the Journal of Clinical Medicine. The early-onset cataract (EOC) subject is a relatively less explored field, even though some previous scientific evidences were found in literature. The aim of this article is to focus on the risk factors for EOC in the Taiwan population. Noteworthy the possible association between EOC and antihistamine use, connected to the chronic oxidative environment in an atopic subject. Unfortunately, due to the small cohorts enrolled, it is complex to draw any substantial data on a real risk factor. Furthermore, the main text included several formal mistakes (EOC not explained in the abstract, several sentences in the results/conclusion too long to be clear, a p value of 0.000 to be revised) which must be addressed before a possible re-submission. 

Author Response

  1. due to the small cohorts enrolled, it is complex to draw any substantial data on a real risk factor.                                                                                            Line 168-169 The sample size of this study was determined using G*Power analysis, under effect size d = 0.5,α = 0.05, power (1-β) = 0.90. The calculated results of the total sample size were 70.
  2. Furthermore, the main text included several formal mistakes (EOC not explained in the abstract                                                                            Revised,  Line 17, 23, 83-86 
  3. several sentences in the results/conclusion too long to be clear            Revised
  4. a p value of 0.000 to be revised) which must be addressed before a possible re-submission.                                                                              Revised

Round 2

Reviewer 1 Report

The data presentation has been improved, however, the English language must be further improved, especially in the discussion

Reviewer 2 Report

Dear Author(s),

Thanks for your re-submission on the JCM. The revisions made completely change the value of the article. I would like to congratulate for the extensive work performed to improve the article.

This manuscript is a resubmission of an earlier submission. The following is a list of the peer review reports and author responses from that submission.